# Cell-Free Hemoglobin Does Not Attenuate the Effects of SARS-CoV-2 Spike Protein S1 Subunit in Pulmonary Endothelial Cells

**DOI:** 10.3390/ijms22169041

**Published:** 2021-08-22

**Authors:** Sirsendu Jana, Michael R. Heaven, Abdu I. Alayash

**Affiliations:** Laboratory of Biochemistry and Vascular Biology, Center for Biologics Evaluation and Research, Food and Drug Administration (FDA), Silver Spring, MD 20993, USA; sirsendu.jana@fda.hhs.gov (S.J.); Michael.Heaven@fda.hhs.gov (M.R.H.)

**Keywords:** COVID-19, spike protein, endothelium, bioenergetics, proteomics

## Abstract

SARS-CoV-2 primarily infects epithelial airway cells that express the host entry receptor angiotensin-converting enzyme 2 (ACE2), which binds to the S1 spike protein on the surface of the virus. To delineate the impact of S1 spike protein interaction with the ACE2 receptor, we incubated the S1 spike protein with human pulmonary arterial endothelial cells (HPAEC). HPAEC treatment with the S1 spike protein caused disruption of endothelial barrier function, increased levels of numerous inflammatory molecules (VCAM-1, ICAM-1, IL-1β, CCL5, CXCL10), elevated mitochondrial reactive oxygen species (ROS), and a mild rise in glycolytic reserve capacity. Because low oxygen tension (hypoxia) is associated with severe cases of COVID-19, we also evaluated treatment with hemoglobin (HbA) as a potential countermeasure in hypoxic and normal oxygen environments in analyses with the S1 spike protein. We found hypoxia downregulated the expression of the ACE2 receptor and increased the critical oxygen homeostatic signaling protein, hypoxia-inducible factor (HIF-1α); however, treatment of the cells with HbA yielded no apparent change in the levels of ACE2 or HIF-1α. Use of quantitative proteomics revealed that S1 spike protein-treated cells have few differentially regulated proteins in hypoxic conditions, consistent with the finding that ACE2 serves as the host viral receptor and is reduced in hypoxia. However, in normoxic conditions, we found perturbed abundance of proteins in signaling pathways related to lysosomes, extracellular matrix receptor interaction, focal adhesion, and pyrimidine metabolism. We conclude that the spike protein alone without the rest of the viral components is sufficient to elicit cell signaling in HPAEC, and that treatment with HbA failed to reverse the vast majority of these spike protein-induced changes.

## 1. Introduction

COVID-19 (i.e., coronavirus disease 2019) is responsible for causing severe acute respiratory syndrome, abbreviated as SARS-CoV-2. This disease results in a wide spectrum of clinical symptoms ranging from asymptomatic infection to acute respiratory distress syndrome, multifunctional organ dysfunction, and death. The SARS-CoV-2 spike protein (S1 protein), is involved in the first critical step in the interactions between the virus and the host cell surface receptors, thus facilitating viral cell entry. The spike protein attaches the virus to its cellular receptor, angiotensin-converting enzyme 2 (ACE2), through a defined receptor-binding domain (RBD) on the S1 protein [1]. Anecdotal evidence further supports this mechanism of viral host cell entry since increased levels of ACE2 have a positive correlation with COVID-19 infection [2].

Recent reports have shown that treatment of cultured human pulmonary artery endothelial cells (HPAEC) with the recombinant SARS-CoV-2 spike protein S1 subunit is capable of inducing cell signaling without the rest of the viral components [3]. Prior reports have also demonstrated that the spike protein alone promotes the angiotensin II type 1 receptor (AT1)-mediated signaling cascade, induces the transcriptional regulatory molecules NF-κB and AP-1/c-Fos via MAPK activation, and increases IL-6 release [4]. On the basis of these observations, we can say that the SARS-CoV-2 spike protein alone may trigger cell signaling events culminating in pulmonary vascular remodeling, as well as possibly other cardiovascular complications in humans [3].

Maintaining adequate levels of oxygen under normal physiological settings is critical for the survival of cells. Low or high oxygen levels leads to increased reactive oxygen species (ROS), and therefore both the delivery (by red blood cells and hemoglobin) as well as the consumption of oxygen (by the mitochondria) are precisely regulated by many different molecular mechanisms as part of oxygen homeostasis [5,6]. An important factor that may impact the nature of the interaction between the viral proteins and the AEC2 receptor is the availability of oxygen. Blood oxygenation levels in patients with severe COVID-19 infection have been reported to be very low, a clinical condition termed hypoxemia. However, severe hypoxemia in patients’ blood does not appear to be correlated with the development of tissue hypoxia in COVID-19 patients, thus suggesting a relationship where hypoxia provides a barrier to the function of the SARS-CoV-2 spike protein [7]. 

Hypoxia-inducible factor (HIF-α), together with its iron containing enzyme, prolyl hydroxylase (PHD), are key elements in tissue responses to hypoxia. Under normal oxygen tension (normoxia), the transcriptional activity of HIF-1α is halted by a process of hydroxylation [8]. At the onset of hypoxia, HIF, which contains two subunits, an α-subunit that quickly degrades in the presence of oxygen (its half-life is less than 5 min in 21% O_2_), and a more stable β-subunit that is translocated to the nucleus where it binds to hypoxia response element, results in the activation of a number of target genes to correct for oxygen deficits [9]. Under hypoxic conditions, HIF1-α induces the expression of ACE1, while the expression of ACE2 is markedly decreased.

SARS-CoV-2 viral proteins also interact with host cell mitochondrial proteins, leading to a loss of membrane integrity and dysfunction in the bioenergetics of the mitochondria. These mitochondrial proteins may also serve as damage-associated molecular pattern (DAMP) molecules activating innate immunity. Mitochondrial dysfunction and subsequent pathogenesis due to multiorgan failure that has been linked to COVID-19 infection [10]. Functional mitochondrial analyses of peripheral blood mononuclear cells (PBMCs) from patients with COVID-19 detected mitochondrial dysfunction, increased glycolysis, and high levels of mitokines [11].

In this study, we investigated the interaction between the recombinant spike protein S1 and HPAEC under normoxic and hypoxic conditions. Specifically, we focused on the relationship between oxygen content in the medium, the effects of adding hemoglobin (HbA) as an oxygen carrier, and the regulation of HIF-1 α as well as mitochondrial function. The S1 protein alone caused intracellular signaling and functional changes that were dependent upon oxygen levels, suggesting a relationship between the S1 protein and ACE2 receptor as well as oxygen-sensing HIF signaling proteins and the mitochondrial respiratory machinery. Notably, these changes caused by the S1 protein occur prior to viral entry and therefore may provide a rationale for therapeutic interventions as well as potential long-term effects from the S1 protein upregulation in current messenger RNA vaccines.

## 2. Results

### 2.1. Spike Protein Treatment and Cell Viability

We selected the range of S1 protein concentrations on the basis of several recent studies [3,12,13,14]. We used an intermediate concentration starting from 5 nM up to 25 nM. To assess whether the S1 protein causes differences in cell viability or cytotoxicity, we incubated HPAEC with the S1 protein at various concentrations (5–25 nM) for up to 12 h and found no changes in the cell viability of HPAEC measured by the trypan blue dye exclusion method (Figure 1A). To confirm this finding, we also analyzed S1 protein-treated cells by a cell viability assay that measures the amount of lactate dehydrogenase (LDH) released in the cell culture medium. Consistent with the trypan blue result, the S1 protein did not cause any significant LDH release, verifying there was no loss in the cell viability (Figure 1B). Furthermore, no differences in cell morphology were observed by analysis with an inverted microscope (data not shown). Since 25 nM concentration is physiologically a fairly high level, we did not assess any higher concentration.

### 2.2. S1 Protein Caused Disruption of Pulmonary Endothelial Cell Permeability

Due to endothelial barrier dysfunction that allows T cell trafficking into the lung tissues autopsied from deceased COVID-19 patients [15] as well as acute lung injury reported in murine models of COVID-19 subjected to S1 protein [16], we next assessed whether the S1 protein without all the remaining viral components is sufficient to cause endothelial barrier disruption by growing HPAEC in a tight monolayer and monitoring for changes in endothelial permeability by measuring the passage of FITC conjugated-dextran (40 kD) through the monolayers [17,18]. A significant rise in dextran passage indicative of cell permeability was observed after 12 h at S1 protein concentrations of 10 nM and 25 nM (Figure 2A). As a negative control, we also carried out this experiment in the presence of an antibody against the ACE2 receptor to determine whether blocking of the receptor had any effect on S1 protein-mediated disruption of the endothelial permeability in HPAEC. The findings in Figure 2B show a partial reduction (up to 40%) in S1 protein-induced loss of monolayer integrity in the HPAEC by 100 nM of anti-ACE2 antibody. Our results thus indicate a role of the S1 protein–ACE2 receptor complex in the S1 protein disruption of endothelial barrier function. 

### 2.3. Hypoxia Downregulated the AcE2 Receptor in the Presence and Absence of Hemoglobin 

Hypoxia is among the most prominent clinical features of SARS-CoV 2 pathophysiology [7]. Therefore, to understand if hypoxia affects the ACE2 receptor and other related cellular signaling pathways, we subjected HPAEC to 12 h of hypoxic (1% oxygen) and normoxic experimental conditions. Additionally, since hemoglobin (Hb) is the main oxygen carrier at the tissue and cellular level, we also incubated HPAEC with HbA (65 µM) and the S1 protein (25 nM). Levels of HIF-1α under normoxic and hypoxic conditions are shown in Figure 3A,B. As expected, there was a steady rise in HIF-1α in all the conditions under hypoxia (Figure 3B,C). HIF-1α upregulation in hypoxia was also supported by the proteomic analysis, revealing several signaling precursors of HIF activation (see Section 2.6). However, we found that HbA alone or when co-incubated with S1 spike protein did not cause any significant change in HIF-1α expression over either untreated control or S1 protein-treated cells (Figure 3C). Next, we assessed the ACE2 receptor level under normoxic and hypoxic environments. Our results shown in Figure 3D,E indicate that the basal normoxic levels of ACE2 receptor in HPAEC were significantly downregulated by hypoxia. Surprisingly, incubation with S1 protein resulted in significant downregulation of ACE2 expression under both normoxia and hypoxia. However, addition of HbA did not cause any change in the ACE2 receptor level in normoxia and also in hypoxia compared to respective untreated control cells (data not shown). 

### 2.4. S1 Protein Triggered Endothelial Activation and Induced a Pro-Inflammatory Response in HPAEC

A significant body of evidence indicates endothelial activation and pro-inflammatory cytokines play a critical role in the pathogenesis of the SARS-CoV-2 infection [19]. Additionally, a potential caveat to using HbA as a countermeasure for COVID-19-induced hypoxia is that hemoglobin has been reported to cause stimulation of IL-6, IL-8, and tumor necrosis factor-α from leukocytes in whole blood [20]. Using ELISA-based methods, we quantified the levels of two major adhesion molecules, e.g., VCAM-1 and ICAM-1 in pulmonary endothelial cells. S1 protein exposure caused a significant rise in both VCAM-1 and ICAM-1, indicating the activation and inflammation of these endothelial cells in culture (Figure 4A,B). No significant differences were observed in HbA and S1 protein-treated cells versus S1 protein-only-treated HPAEC, and HbA treatment alone compared to control caused an elevation in the levels of ICAM-1 and VCAM-1 but lacked statistical significance.

Next, we measured several key cytokines to further assess the pro-inflammatory response elicited by the S1 protein for IL-1β, CCL5, and CXCL10. As shown in Figure 4D,E, both CCL5 and CXCL10 were significantly elevated by the S1 protein. We also observed a mild increase in the level of IL-1β by the S1 spike protein (Figure 4C), but the change was not statistically significant. HbA treatment alone versus control caused an insignificant increase in all three cytokines, whereas coincubation with HbA and S1 protein versus S1 protein alone caused significant suppression of CCL5 and CXCL10.

### 2.5. S1 Protein Caused Elevated Mitochondrial ROS and Mild Bioenergetic Impairment

Recent studies have highlighted the involvement of metabolic dysfunction, especially mitochondrial aerobic respiration in various cell types including endothelial cells, possibly contributing to deterioration of pulmonary function and airway hypoxia in COVID-19 patients [11]. Therefore, we monitored mitochondrial superoxide generation (O_2_^−^) as an indicator of ROS production (Figure 5A–C). A significant rise in ROS was detected in S1 protein treated versus untreated cells, as well as in comparisons between the positive control rotenone (a specific complex I inhibitor) versus untreated cells. 

Next, we used the extracellular flux analyzer (XF Assay) to assess the energy utilization in HPAEC exposed to S1 protein and monitored mitochondrial bioenergetics and glycolytic proton flux in real time. The oxygen consumption rate (OCR) and extra-cellular acidification rate (ECAR) in HPAEC were obtained from the XF assays as indicators of mitochondrial respiration and cellular glycolytic activity, respectively. Exposure to S1 protein and HbA did not reveal any significant changes in basal OCR before inhibition of oxidative phosphorylation by oligomycin. When protonophore-induced (FCCP) maximal capacity of the mitochondrial electron transport system (ETS) was achieved, there was also no inhibition seen in S1-treated cells (Figure 5D,E). We also measured glycolytic rates in HPAEC under the influence of S1 protein and HbA treatment with similar experimental conditions. Basal glycolysis was mostly unaffected by either S1 protein or HbA. However, glycolytic capacity (the difference between maximum glycolysis and basal glycolysis) was mildly increased by S1 spike protein (Figure 5F,G), which has previously been reported in peripheral mononuclear blood cells in COVID-19 patients [11].

### 2.6. Proteomic Alterations Related to Oxygen Deprivation in HPAEC 

To further characterize the hypoxic environment that was generated in this study, we compared HPAEC subjected to hypoxia versus normoxia without any other additives (*n* = 3 per condition). As shown in the graphical volcano plot (Figure 6 representing all 2737 quantifiable proteins listed in Appendix A), there were 86 proteins significantly increased and 27 proteins significantly decreased in the hypoxic/normoxic proteome (*p*-value < 0.05). Because a hallmark signature of hypoxia is the activation of the hypoxia-inducible factor (HIF-1α) signaling cascade, we next searched our dataset for HIF-1α, but due to the relatively low abundance of HIFs in the HPAEC proteome, we were not able to detect any HIF isoforms by the LC–MS/MS method employed. Nevertheless, we found increases in egl nine homolog 1 (EGLN1; Figure 6, blue highlighted protein), a key regulator that catalyzes the post-translational modification (PTM) 4-hydroxyproline in the oxygen-dependent degradation domains of HIF [21]. EGLN1 has also been consistently increased in numerous studies involving hypoxia [22,23,24]. Other signs of HIF activation in the hypoxia were indicated by elevations in prolyl 4-hydroxylase subunits (P4HA1, P4HA2; Figure 6, green highlighted proteins) involved in HIF-mediated extracellular matrix (ECM) remodeling in hypoxic conditions by catalyzing the formation of 4-hydroxyproline [25]. As a result, we subsequently searched our proteomic data to detect hydroxylated proline residues (Appendix A). In total, we found 69 differentially abundant 4-hydroxyproline modified peptides with *p*-values < 0.05. Of these, 51/69 (74%) were increased in hypoxic/normoxic HPAEC, and most lacked the EGLN1 recognition motif LXXLAP [26], indicating their modification occurred due to the activity of P4HA1 and P4HA2. Last, we searched the data for ubiquitinylation and observed no peptides with this modification containing significantly altered abundance in comparisons of HPAEC in normoxic versus hypoxic conditions (data not shown).

We next input the list of up- and downregulated proteins (*p*-value < 0.05) in hypoxia versus normoxia into the database for annotation, visualization, and identification using the DAVID pathway analysis tool. Using the list of increased proteins in hypoxia relative to normoxia, we identified the ECM–receptor interaction in the Kyoto Encyclopedia of Genes and Genomes (KEGG) pathway with several proteins known to be elevated in hypoxia [27] (von Willebrand factor, basement membrane-specific heparan sulfate proteoglycan core protein, thrombospondin-1, fibronectin; Figure 1, red highlighted proteins). No other significant DAVID pathway results were obtained for up- or downregulated proteins. Due to the numerous hypoxia-related signaling including increases in HIF activated proteins (EGLN1, P4HA1, P4HA2) [22,23,24,25], remodeling of the ECM–receptor interaction pathway [27], and our immunoblots of HIF-1α (Figure 3A,B), these findings support the conclusion that the experimental conditions for oxygen deprivation faithfully created a hypoxic phenotype in the current study.

### 2.7. Quantitative Proteomic Analysis of the S1 Protein in Hypoxia and Normoxia

To characterize the proteins altered by the S1 protein in HPAEC, we compared S1 protein-treated versus non-treated cells. In normoxic conditions, there were 136 proteins significantly increased and 28 proteins significantly decreased in S1 protein treated versus untreated HPAEC (Appendix A). Using the DAVID pathway tool indicated the S1 protein in normoxic conditions caused upregulation of proteins in the lysosome (PSAP, AGA, TCIRG1, CTSA, CTSZ, LIPA, SCARB2, TPP1, GALNS, NAGA, GAA), ECM–receptor interaction (COL4A1, COL5A1, VWF, HSPG2, THBS1, LAMA4, FN1), RNA degradation (CNOT7, NUDT16, DCPS, ENO2, RQCD1), and focal adhesion (JUN, COL4A1, COL5A1, VWF, THBS1, LAMA4, FN1) KEGG pathways. On the other hand, for the 28 proteins significantly decreased by the S1 protein in normoxic conditions, we found enrichment for the pyrimidine metabolism (CAD, RRM2, TYMS) and metabolic (CAD, RRM2, CYP51A1, GLUD1, TYMS, NDUFA10) KEGG pathways. In contrast, in a hypoxic environment, only 21 proteins were significantly increased and 18 were significantly decreased in S1 protein versus non-treated HPAEC during hypoxia (Appendix A), and none of the altered proteins were significantly associated with a particular KEGG pathway.

To directly interrogate the mechanism responsible for attenuating the S1 protein response in hypoxia, we compared hypoxic cells treated with the S1 protein to normoxic cells that were also treated with the S1 protein (Appendix A). We found 49 significantly elevated and 65 significantly decreased proteins in the hypoxic/normoxic HPAEC in the presence of the S1 protein. With the list of 49 proteins elevated, we observed a significant association using DAVID for the Parkinson’s disease (HTRA2, SDHA, NDUFS5, NDUFA10) and metabolic (SDHA, PIK3C3, RRM2, GNPDA2, CYP51A1, NDUFS5, TYMS, IMPA1, NDUFA10, PDXK) KEGG pathways. Additionally, we searched the list of 65 proteins with lower abundance in the hypoxic/normoxic comparison in the presence of the S1 protein, which identified the endocytosis (RAB4A, AP2S1, HLA-B, CAV2, VPS25, SNF8) and epithelial cell signaling in Helicobacter pylori infection (JUN, PLCG1, MAPK9) KEGG pathways. Since endocytosis is necessary for SARS-CoV-2 cell entry [28], the reduction of proteins in this pathway is consistent with ACE2 receptor downregulation in hypoxia relative to normoxia (Figure 3A,B).

### 2.8. Protein Differences Caused by HbA in the Presence of the S1 Protein

Since COVID-19 patients exhibit hypoxemia during disease progression [29], we next assessed the possible therapeutic potential of adding HbA as an oxygen carrier. By comparing the hypoxic proteomes from S1 protein- and HbA-treated cells versus S1 protein-only-treated cells, we were able to quantify 2723 proteins with 20 significantly increased proteins and 73 significantly decreased proteins (Appendix A); neither the list of increased proteins nor the list of decreased proteins were associated with any KEGG pathways. In normoxic conditions, the comparison of cells treated with the S1 protein and HbA versus the S1 protein only resulted in the quantification of 3112 proteins (Appendix A). There were 22 proteins significantly increased and 32 proteins significantly decreased (*p*-value < 0.05). From inputting the list of increased proteins into the DAVID pathway analysis program, we found no significant KEGG pathways. However, from analysis of the list of decreased proteins, we identified the focal adhesion KEGG pathway (JUN, COL4A1, CRKL, COL5A1). Representative peptide LC–MS/MS chromatograms showing the relative quantification of these proteins reduced in the focal adhesion pathway are shown in Figure 7.

## 3. Discussion

In this report, we hypothesized that subjecting pulmonary endothelial cells treated with the S1 spike glycoprotein to cell-free Hb may attenuate the SARS-CoV-2 virus pathophysiology, on the basis of the rationale that the S1 protein may bind Hb [30] and the presence of an oxygen carrier may provide therapeutic benefits to COVID-19 patients exhibiting hypoxia [7,31]. However, the potential negative aspects of cell-free Hb relevant to the SARS-CoV-2 infection process were also evaluated, including whether Hb induces pro-inflammatory cytokines [20]. Additionally, since hypoxia has been reported to deplete the levels of the cell surface receptor ACE2 for the viral entry [32], we tested the effects of the S1 protein in normoxia and hypoxia by measuring the levels of ACE2 and HIF-1α in the presence and absence of the oxygen carrier human HbA. Last, to determine altered cell signaling pathways, we employed quantitative proteomics to analyze the effects of the S1 protein and HbA in both hypoxic and normoxic conditions.

The entry of the SARS-CoV-2 virus inside host cells is facilitated by ACE1 and ACE2 expressed in the lungs, kidneys, heart, and arteries [1]. In our results shown in Figure 2, it was evident that the ACE2 receptor was almost undetectable in hypoxia, and as expected, the ACE2 level in normoxia was vastly higher. As anticipated, the levels of HIF-1α had an inverse relationship to ACE2, where HIF-1α was higher in hypoxic compared to normoxic experimental conditions. Moreover, we also found that presence of HbA had no effect on either HIF-1α expression or on the levels of the ACE2 receptor. A similar phenomenon was also reported by some other groups where hypoxia was shown to downregulate spike protein-binding heparan sulfate and heparan sulfate containing proteoglycans along with ACE2 receptors [13,33] Collectively, the observed dependence on normal oxygen tension for maintaining ACE2 receptor level also fits with the observation that less severe COVID-19 cases were seen among a high-altitude Tibetan population [34], although many other factors may also have contributed to these favorable patient outcomes. Unfortunately, most COVID-19 patients during the initial phases of infection have normal oxygen tension, allowing for substantial amounts of viral entry due to sufficient amounts of the ACE2 receptor.

In subsequent stages of the infection process, immune cells are trafficked into the lungs, and it has been suggested that this occurs in part due to a loss in pulmonary endothelial cell barrier function [15]. Therefore, we assessed the endothelial barrier function in the HPAEC analyzed in the presence of the S1 protein and found significantly higher cell permeability in cells treated with S1 protein compared to untreated control cells. This finding confirms recent work by Biancatelli et al. [16], indicating the S1 protein alone is capable of inducing barrier dysfunction in human pulmonary microvascular cells and is notable given that HPAEC represent a substantial amount of all lung endothelial alveolar cells and provide a possible mechanistic basis for understanding why the lungs are a target organ for immune-mediated destruction in COVID-19 patients [35]. In addition to the endothelial cell barrier dysfunction, the process of infection also involves elevations in pro-inflammatory cytokines to such a degree it has been referred to as a “cytokine storm” with IL-6 serving as the upstream inducer of nuclear factor kappa B (NF-κB) and signal transducer and activator of transcription 3 (STAT3) [36]. In this study, because we initially found evidence that the cell barrier function of the pulmonary arterial endothelial cells is compromised by the S1 spike protein alone (Figure 4A,B), we next determined whether VCAM-1 and ICAM-1 were elevated since these proteins have a critical role in mediating the interaction between leukocytes and endothelial cells in inflammatory conditions [37]. Once activated, endothelial cells upregulate expression of cell adhesion molecules (CAMs) and pro-inflammatory cytokines that play a key initial role in the process of inflammation. Both VCAM-1 and ICAM-1 were significantly elevated by the S1 spike protein and attempts to lower the levels with HbA failed to significantly alter the levels of these inflammatory molecules. As a result, we also analyzed whether the S1 spike protein alone would cause other cytokines to be elevated, including IL-1β, which has been proposed as a therapeutic target [38], as well as the inflammatory chemokines CCL5 and CXCL10. All of these cytokines were highly elevated in comparison between the S1 spike protein and untreated cells, and notably for CCL5 and CXCL10, the treatment with HbA significantly lowered the levels of these cytokines but caused an insignificant rise in the level of IL-1β (Figure 4C–E). With respect to the effect of HbA alone versus untreated cells, as it relates to the potential for negative side effects as a countermeasure treatment for COVID-19, we consistently observed an increase in every cytokine evaluated in the study, confirming that this is a necessary readout to evaluate the effects of cell-free Hb [20].

Since the entire viral package encompassed by the SARS-CoV-2 infection hijacks the mitochondria of immune cells, causing altered bioenergetics leading to cell death [39], an initial parameter evaluated in this study was whether the S1 spike protein alone could alter cell viability (Figure 1). Data obtained from two methods for evaluating cell viability indicated the S1 spike protein alone is not capable of mediating significant differences, suggesting other components of the virus are necessary to induce cell death. However, loss of endothelial monolayer integrity mediated by the S1 protein observed in our experimental setup is consistent with a recent observation where a similar loss of endothelial barrier function by spike protein was shown using an advanced 3D microfluid model of the human blood–brain barrier [12]. This group also found similar pro-inflammatory cytokine responses elicited by the spike protein that were considered a contributing factor for losses in membrane integrity [12]. Our data also clearly shows that cell adhesion molecules ICAM-1 and VCAM-1, which are key to the trans-endothelial migration of immune cells following inflammatory challenge, were upregulated by S1 protein and in conjunction with reduced barrier integrity. Both these phenomena can be an outcome of observed cytokine expression since pro-inflammatory cytokines are known to regulate tight junction protein degradation [40]. It is important to note that ACE2 is also known to reduce endothelial inflammation [38]. In contrast, ACE2 has been shown to inhibit endothelial proliferation, whereas we have shown here that ACE2 is downregulated under hypoxia, which actually can promote endothelial survival [41]. 

Energetically, endothelial cells are heavily dependent on anerobic metabolism from the glycolytic pathway, but mitochondrial homeostasis also plays a crucial role in endothelial cell signaling apart from contributing to ATP generation [42,43]. Recently, a study revealed endothelial cells treated with S1 protein caused increased mitochondrial fragmentation, indicating altered mitochondrial dynamics [13]. In order to identify any impact of S1 spike protein on the endothelial cellular bioenergetics, we first analyzed mitochondrial superoxide production and found the spike protein alone can induce some degree of ROS production from endothelial mitochondria compared to untreated controls (Figure 5A–C). Although we could not see any noticeable impact of S1 protein on the actual oxygen consumption by the actively respiring mitochondria, a concomitant rise in glycolytic reserve capacity in those cells indicates a possible compensatory effect of subtle dysfunctional electron transport chain. Therefore, it is also possible that the observed mitochondrial ROS were primarily generated from the impaired electron transport through the respiratory chain complexes as a function of decreasing oxygen tension that can also in turn inhibit HIF-1*a* prolyl hydroxylation and degradation [44]. Endothelial mitochondria can actually act as sensor for oxygen in addition to nitric oxide for the vasodilatory response and therefore can play a “reconnaissance” role through relaying information to adjacent cardiac myocytes or smooth muscle cells [42]. Therefore, mitochondrial electron transport and subsequent oxidative phosphorylation are interconnected as signaling mechanisms conveying cellular oxygen availability to oxygen-sensing pathways [5]. However, further analysis of mitochondrial metabolism alterations caused by the S1 spike protein alone indicated a mild increase in glycolytic capacity and no difference in mitochondrial oxygen consumption rates (Figure 5D–G). Additionally, treatment with HbA also did not significantly impact glycolytic capacity or mitochondrial oxygen consumption.

The untargeted quantitative proteomic analysis of untreated cells in hypoxic versus normoxic conditions resulted in verifying an array of proteins related to HIF signaling in hypoxia including EGLN1 [21,22,23,24] and P4HA1, as well as P4HA2, which is involved in ECM remodeling in hypoxia by catalyzing the formation of 4-hydroxyproline [25] (Figure 6). However, our quantitative proteomic analysis indicated very few proteins were altered in hypoxic cells treated with S1 spike protein versus untreated cells. In contrast, numerous cell-signaling pathways were altered in normoxic conditions comparing S1 spike protein-treated versus -untreated cells including upregulation of lysosome, ECM-receptor interaction, RNA degradation, and focal adhesion KEGG pathways and downregulation of pyrimidine metabolism and metabolic KEGG pathways. A plausible explanation for the lack of protein differences in the hypoxic cells treated with the S1 spike protein is that the ACE2 receptor downregulation in hypoxia (Figure 3) prevents its downstream effects and ability to interact with the endothelial cells analyzed [45]. Notably, in comparisons of hypoxic S1 spike protein versus normoxic S1 spike protein-treated cells, we found elevated proteins were significantly associated with the Parkinson’s disease KEGG pathway. However, unlike Parkinson’s disease, where the activity of proteins in this pathway are decreased, causing mitochondrial dysfunction due to mutations or other factors [46], in the current study, these proteins were increased, suggesting a mechanism of mitochondrial protection induced in the hypoxic cells treated with S1 protein. Last, we assessed what impact adding HbA might have upon the proteomic response induced by the S1 spike protein. In normoxic HPAEC treated with HbA and S1 protein versus S1 protein-only-treated cells, we identified significant downregulation of the focal adhesion KEGG pathway (Figure 7). This finding may be particularly relevant to the SARS-CoV-2 infection process, given that ACE2 is required for viral entry [47], and the ectodomain of ACE2 activates focal adhesion [48].

In many ways, the cumulative findings in this report do not support the hypothesis that HbA would be an effective countermeasure to use for COVID-19 patients, specifically because the addition of HbA was unable to attenuate the levels of numerous cytokines (Figure 4) or significantly alter mitochondrial bioenergetics (Figure 5). However, an important limitation with this finding is that throughout the current study, only the S1 subunit of the spike protein was used rather than live SARS-CoV-2 virus; HPAECs were treated with HbA instead of in vivo models of disease; and HbA, which readily loses heme, was evaluated rather than other Hb oxygen carriers that are less susceptible to heme loss. The use of only one viral protein instead of infectious SARS-CoV-2 virus is a limitation to this study, and also endothelial cells are not ideal for active viral replication [49]; however, our data clearly reveal that S1 protein alone can damage pulmonary endothelial cells manifested by impaired mitochondrial function and increased glycolysis coupled with pro-inflammatory cytokine release, which overall can lead to endothelial activation and loss of barrier function.

## 4. Materials and Methods

### 4.1. Preparation of Hemoglobin from Blood 

Human Hb was isolated from erythrocyte lysates by anion DEAE and cation CM chromatography, respectively, as reported previously [50]. Following isolation, Hb was concentrated, dialyzed against PBS, and was stored at −80 °C for future use. The criteria of purity of HbA protein was verified by isoelectric focusing and HPLC. HbA solution was also passed through an endotoxin removal column (Thermo Fisher Scientific, Waltham, MA, USA) to remove LPS, and final concentration of Hb solution was calculated as reported previously [51].

### 4.2. Endothelial Cell Culture

Cryopreserved human pulmonary arterial endothelial cells (HPAEC) (Thermo Fisher Scientific, Waltham, MA, USA) were cultured in a specially formulated media (Medium 200) containing 2% fetal bovine serum (FBS) supplemented with Low Serum Growth Supplement (LSGS) (Thermo Fisher Scientific, Waltham, MA, USA). For all experiments, HPAEC were used between passages 5 and 10. 

### 4.3. Treatment of HPAECs with S1 Spike Protein

HPAECs were grown to 80–90% confluency in complete media before any treatments. Cells were then serum starved for 12 h in FBS-free medium composed of Medium 200 supplemented with all other components of LSGS Kit except FBS (Thermo Fisher Scientific, Waltham, MA, USA). LPS-free recombinant SARS-CoV-2 Spike Protein S1 (aa 1–681, catalog no. AGX818) was purchased from Sigma-Aldrich (St. Louis, MO, USA). Cells were exposed to either S1 protein alone or with HbA (65 µM) for up to 24 h in serum-free growth media. After the incubation, cells were washed with pre-warmed phosphate-buffered saline (PBS) three times to remove any residual S1 or Hb proteins in the media. Cells were then lysed with RIPA lysis and extraction buffer (Thermo Fisher Scientific, Waltham, MA, USA) containing protease inhibitor for further studies. 

### 4.4. Gel Electrophoresis and Immunoblotting 

HPAEC lysate proteins were resolved by SDS-PAGE using precast 4–20% NuPAGE bis-tris gels (Thermo Fisher Scientific, Waltham, MA, USA) and then transferred to nitrocellulose membranes (BioRad, Hercules, CA, USA) by standard immunoblotting technique. Nitrocellulose membranes were processed with different specific primary antibodies and appropriate secondary antibodies. Monoclonal antibodies against HIF-1α (rabbit, ab51608), ACE2 (rabbit, ab108252), and actin (mouse, ab8226) were purchased from Abcam (Cambridge, MA, USA). Appropriate HRP-conjugated goat anti-mouse IgG (ab97040) and anti-rabbit IgG (ab205718) secondary antibodies were also obtained from Abcam (Cambridge, MA, USA). Blots were then probed with SuperSignal™ West Pico PLUS Chemiluminescent Substrate (Thermo Fisher Scientific, Waltham, MA, USA). 

### 4.5. Transwell Assay to Measure HPAEC Monolayer Integrity

For permeability determinations, HPAEC were grown to confluence on collagen-coated PTFE-Transwell membrane inserts (Corning Inc., Corning, NY, USA) placed on a 24-well culture plate forming a two-compartment system. Approximately 2 × 105 cells in 0.1 mL of complete medium were seeded onto the membrane insert with pore size of 0.4 µm following a previously published method [18]. The lower compartment was then supplemented with 0.6 mL of complete media to equilibrate the hydrostatic fluid pressures across the membranes per the manufacturer’s protocol. Cells were grown up to 5–6 days until they formed a tight confluent monolayer.

To determine the macromolecular passage across the membranes, we added 0.1 mL of FITC-labelled dextran, 40 KD (200 µg/mL) solution prepared in complete media with or without Hbs, and other additives onto the membrane insert following complete removal of culture medium. Simultaneously, each well in the lower chamber was replenished with 0.6 mL of fresh culture medium. Diffusion of dextran was monitored after varying periods of incubation by measuring the FITC-green fluorescence in the medium in the lower chamber using a Synergy HTX Multi-Mode Reader (Biotek Instruments, Inc., Winooski, VT, USA).

### 4.6. Measurement of Endothelial Activation and Pro-Inflammatory Cytokines

Endothelial activation was assessed by measurement of VCAM-1 and ICAM-1 utilizing ELISA-based commercial assay kits purchased from Thermo Fisher Scientific (Waltham, MA, USA). Pro-inflammatory cytokines, e.g., IL-1β, CCL5 (RANTES), and CXCL10 were also measured by ELISA-based assays obtained from Thermo Fisher Scientific (Waltham, MA, USA).

### 4.7. Mitochondrial Superoxide Measurement and Fluorescence Microscopy

Mitochondrial superoxide generation was measured in the HPAEC following exposure to the S1 protein using MitoSOX Red, which is a specifically targeted fluorescent probe to mitochondria in live cells (Thermo Fisher Scientific, Waltham, MA, USA). To detect mitochondrial superoxide in HPAEC following exposure to S1 protein, we washed both S1 treated and untreated control cells grown in 96 wells once with warm PBS. Then, cells were loaded with the MitoSOX probe (5 µM) prepared in Hank’s balanced salt solution with calcium and magnesium (HBSS/Ca/Mg). After 15 min of incubation at 37 °C, cells were gently washed with warm HBSS, and the red fluorescence (Ex 510 nm, Em 590 nm) was monitored using a Synergy HTX Multi-Mode Reader (Biotek Instruments, Inc., Winooski, VT, USA). A few positive control wells were kept by exposing untreated cells in those wells to mitochondrial complex I inhibitor rotenone (5 µM) for 1 h prior to the MitoSOX dye addition.

In a separate set of experiments, cells were grown on coverslips up to 50% confluency and exposed to S1 protein for 12 h. Treated and untreated cells were probed with MitoSOX dye, as described above. Cells were visualized under an EVOS fluorescence microscope (Thermo Fisher Scientific, Waltham, MA, USA) after staining with DAPI nuclear dye (0.2 µM).

### 4.8. Endothelial Bioenergetic and Glycolytic Flux Measurements

To monitor endothelial oxygen consumption and glycolytic rate in real time, we used an Agilent-Seahorse XF24 Extracellular Flux analyzer (Agilent, Santa Clara, CA, USA), as described previously [17]. Briefly, 100,000 cells were added to each well of a collagen I-coated 24-well XF-V7 cell culture plate (Agilent, Santa Clara, CA, USA) and were cultured overnight. Cells were then serum-starved for 12 h prior to incubation with S1 protein alone or with HbA (65 µM) for another 24 h. Following the incubation, media was gently washed once with PBS to remove S1 protein or HbA and replaced with 500 µL of XF-assay media (Agilent, Santa Clara, CA, USA) supplemented with 10 mM glucose, 5 mM pyruvate, and 2 mM glutamate. Mitochondrial oxygen consumption rate (OCR) was assessed following automated sequential injections of oligomycin (1 µM), carbonyl cyanide-p-trifluoro-methoxyphenylhydrazone (FCCP, 1 µM), and a combination of mitochondrial inhibitors (rotenone, 1 µM, and antimycin A, 1 µM) that created different bioenergetic states, e.g., coupled, uncoupled, and complete inhibition, respectively [17]. In a similar set of experiments, endothelial glycolytic capacity was assessed by measuring the extracellular acidification rate (ECAR) where real-time glycolytic profile was obtained by sequential addition of glucose (10 mM), oligomycin (1 µM), and glycolytic inhibitor 2-deoxyglucose (2-DG, 100 mM) to the wells. A glucose-free XF-assay media was used for ECAR experiments instead of standard XF assay media. 

Following the completion of XF assay, the OCR and ECAR plots were generated and analyzed by the XF24 onboard software (version 1.8) (Agilent, Santa Clara, CA, USA). Average OCR or ECAR values from blank wells were subtracted to eliminate any background. Various bioenergetic and glycolytic parameters were calculated as described previously [52]. We calculated basal respiration by subtracting rotenone/antimycin inhibited (non-mitochondrial) from the initial OCR before addition of oligomycin, whereas maximum OCR was obtained by the difference between maximum OCR achieved by FCCP and non-mitochondrial OCR. Similarly, ECAR achieved after addition of glucose was considered as basal glycolysis, and maximum ECAR achieved by oligomycin by shutting down the ATP generation through oxidative phosphorylation represented the glycolytic capacity. 

### 4.9. Micro Data-Independent Acquisition (µDIA) Mass Spectrometry Sample Preparation and Acquisition

An amount corresponding to 10 µg of total protein estimated by the BCA assay from HPAEC lysates from each sample were subjected to S-trap mini column (Protifi, Farmingdale, NY, USA) trypsin digestion with iodoacetamide alkylation, according to the manufacturer’s instructions. *n* = 3 per condition was analyzed in all proteomic experiments. Digested peptides from each sample were analyzed by LC–MS/MS on an Exploris 480 Orbitrap mass spectrometer (Thermo Scientific, Waltham, MA, USA) in conjunction with an UltiMate 3000 RSLCnano UHPLC and an EASY-Spray nanoelectrospray ionization source operating in positive ion mode. Peptides were loaded on an Acclaim PepMap100 (75 µm × 20 mm, 3 µm) C18 reverse phase trap before being separated using an EASY-Spray 75 µm × 250 mm 2 µm C18 reverse phase analytical column (Thermo Scientific). Peptides were eluted with an increasing percentage of acetonitrile over a 120 min gradient with a flow rate of 300 nL/minute at 40 °C. The mass spectrometer was operated in data-independent acquisition (DIA) mode with MS2 spectra acquired at 77 distinct *m*/*z* 8 shifted mass windows stepping from *m*/*z* 400–1008 with a scan range of *m*/*z* 200–1500, and an MS survey scan was obtained once every duty cycle from *m*/*z* 400–1020. MS and MS2 spectra were acquired with an Orbitrap scan resolution of 240,000 and 30,000, respectively. The accumulation gain control (AGC) was set to 1 × 10^6^ and 5 × 10^5^ ions, with maximum injection times of 80 and 40 ms for MS and MS2 scans, respectively. The precursor ions in MS2 fragment ion scans were selected across a mass isolation window of *m*/*z* 12 and dissociated by HCD (High Energy Collisional Dissociation) with a 30% normalized collision energy.

### 4.10. Discovery µDIA Proteomic Data Analysis

Raw mass spectra were analyzed using Protalizer µDIA software (v1.1.3.2) from Vulcan Biosciences (Birmingham, AL, USA) [53]. Peptide and protein identifications were made using the Protein Farmer search engine against the canonical and isoform forward and reversed human Swiss-Prot database (2020–10 release). Mass spectra were identified and quantified with a 10 ppm parent and fragment ion mass tolerance after mass calibration. Carbamidomethylation of cysteine residues was searched as a fixed modification in all analyses, and 4-hydroxyproline and ubiquitinylation of lysine were specifically indicated in the results. Peptides with one trypsin missed cleavage were included in the analysis. A strict false discovery rate (FDR) based on a reversed database search of <1% at the peptide level and 1% at the protein level was applied for each sample analyzed. Normalized peptide and protein quantitative relative abundance values were calculated and subjected to a two-tailed unpaired *t*-test as described previously [53].

### 4.11. Statistical Analysis

Plotting of data and statistical calculations were conducted with GraphPad Prism 7 software. All values are expressed as mean ± SD. A *p*-value of <0.05 was considered statistically significant. The difference between two means were compared using unpaired Student’s *t*-test. All error bars in the bar diagrams are indicative of SD.

## Figures and Tables

**Figure 1 ijms-22-09041-f001:**
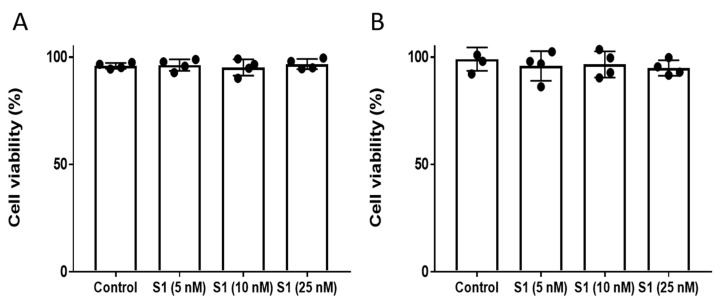
S1 spike protein had no effect on endothelial cell viability. HPAEC were exposed to varying concentrations (5–25 nM) of S1 spike protein for 12 h. Cell viability was assessed by (**A**) trypan blue dye-exclusion method or by (**B**) a cytotoxicity assay based on lactate release. Values are expressed as percentage of live cells (*n* = 4).

**Figure 2 ijms-22-09041-f002:**
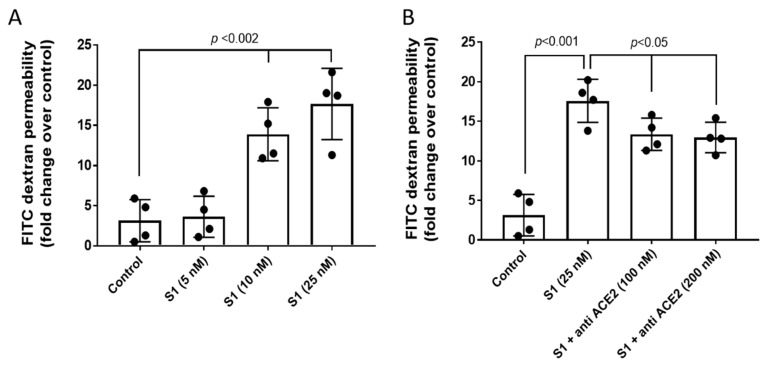
S1 spike protein alone caused disruption of the pulmonary endothelial cell monolayer. (**A**) HPAEC were grown on Transwell inserts to form uniform monolayer and then were exposed to various concentrations of S1 spike protein (5–25 nM) for 12 h in the presence of 40 kD-FITC conjugated dextran. Endothelial monolayer integrity was assessed by measuring the passage of FITC-dextran molecule through the monolayer. FITC-green fluorescence was monitored by fluorescence plate reader as described in Section 4. (**B**) In a similar set of experiments, HPAEC were co-incubated with an anti-ACE2 antibody (100 nM) and S1 spike protein (25 nM) for 12 h in the presence of 40 kD-FITC conjugated dextran. Values are presented as average fold difference over control. (*n* = 4). Bars represent average mean value, each dot in the bars represent individual data points and vertical error bars represent SD. Unpaired Student’s *t*-test. Statistical significance between groups were indicated by brackets with *p* values.

**Figure 3 ijms-22-09041-f003:**
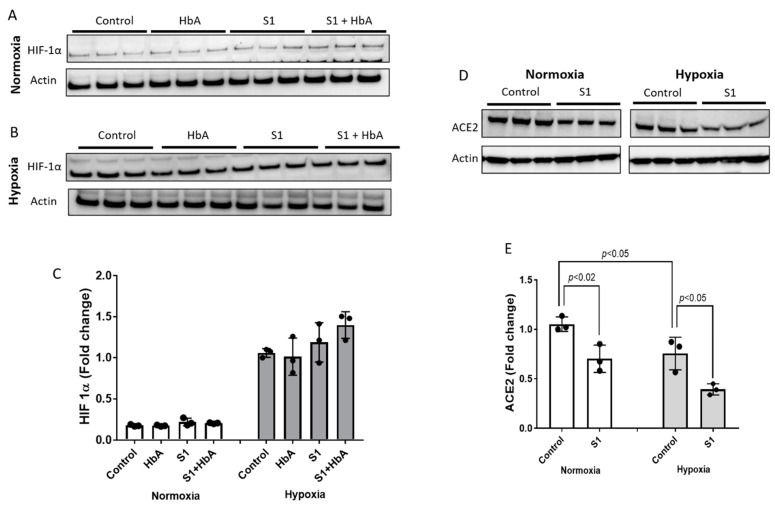
ACE2 receptors were downregulated by hypoxia in HPAEC. Cells were grown to confluence and then either subjected to hypoxia (1% oxygen) (**B**,**D**) or kept at normoxic conditions (**A**,**D**) for 12 h with or without S1 spike protein (25 nM) and HbA (65 µM). Following incubation, cell lysates were immunoblotted with primary antibodies against HIF-1α and ACE2 (**C**,**E**). Equal loading was confirmed by re-probing the blots against actin. Densitometric analysis of all immunoblots were performed, and protein levels were expressed as fold change over respective controls. All immunoblot panels shown are representatives of three separate independent experiments. Values shown in the bar diagrams are means of band intensities (*n* = 3) from the corresponding immunoblots, and vertical error bars represent SD. Statistical significance between groups were indicated by brackets with *p* values, unpaired Student’s *t*-test.

**Figure 4 ijms-22-09041-f004:**
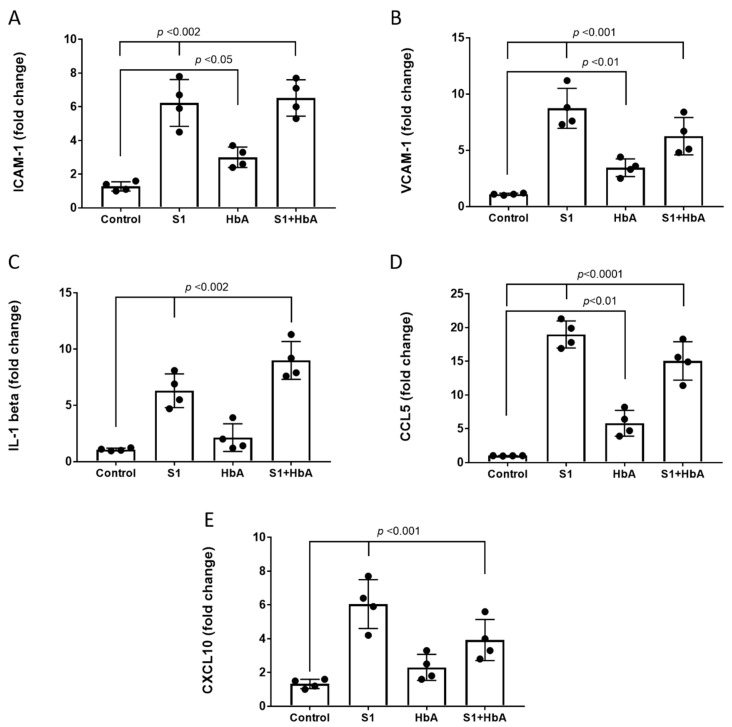
S1 protein induced pulmonary endothelial activation and a pro-inflammatory response. HPAEC were incubated with S1 spike protein (25 nM) with or without HbA for 24 h. Following incubation and addition of S1 protein or HbA where indicated, the cells were lysed and the endothelial adhesion molecules (**A**) ICAM-1 and (**B**) VCAM-1 levels were measured by ELISA. Three major cytokines, e.g., (**C**) IL-1β, (**D**) CCL5, and (**E**) CXCL10 levels were also monitored by cytokine assay kits as described in Section 4. All bars represent average mean values as fold change over respective controls, each dot in the bars represent individual data points and vertical error bars represent SD (*n* = 4). whereas for IL-1 beta, CCL5, and CXCL10 (**C**–**E**), the values were expressed as per milligram protein. Unpaired Student’s *t*-test, statistical significance between groups were indicated by brackets with *p* values.

**Figure 5 ijms-22-09041-f005:**
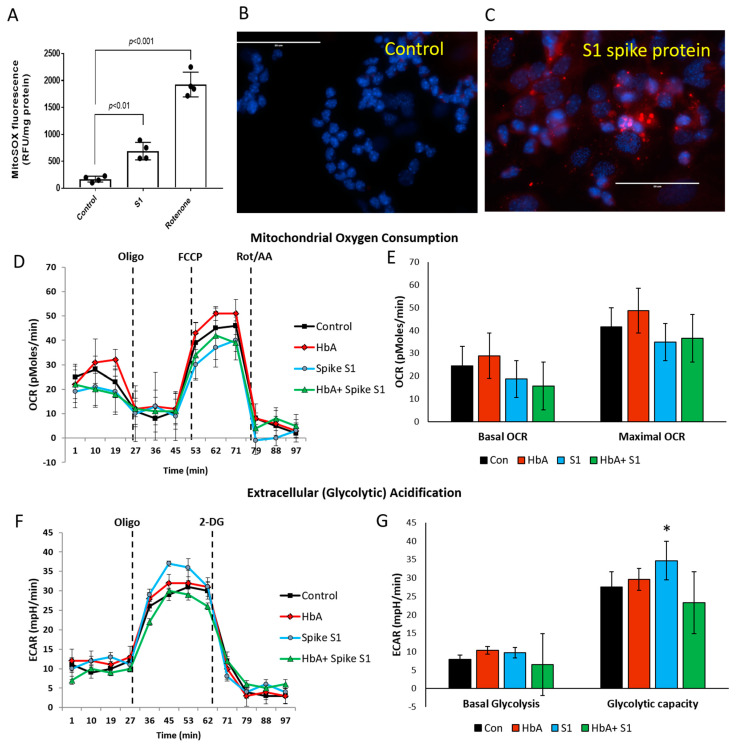
S1 protein increased mitochondrial ROS with only mild effects on pulmonary endothelial cell bioenergetics. HPAEC were exposed to S1 spike protein (25 nM) for 24 h. Mitochondrial superoxide production was assessed by MitoSOX fluorescence probe either by using a fluorescence plate reader (*n* = 4) (**A**) or visualized under a fluorescence microscope (**B**,**C**). Rotenone, a specific complex I inhibitor, was kept as a positive control. All bars represent average mean values, each dot in the bars represent individual data points and vertical error bars represent SD, Student’s *t*-test unpaired. Statistical significance between groups were indicated by brackets with *p* values. White solid lines in (**B**,**C**) indicate length of 50µm. Images are representative of three identical experiments. Mitochondrial bioenergetics was assessed in real time by an Agilent Seahorse (XF24) extracellular flux analyzer. (**D**) Mitochondrial aerobic respiration was measured as oxygen consumption rate (OCR) and (**E**) bioenergetic profile indicating average OCR values from four similar wells treated with S1 spike protein (25 nM) with or without HbA (65 µM). Basal and maximal respiration (OCR) were calculated from the OCR plots (*n* = 4). (**F**) Glycolytic lactate production was measured as ECAR and plotted as the average of four similar wells treated with S1 spike protein (25 nM) with or without HbA. (**G**) Basal glycolysis rate and glycolytic capacity were calculated from ECAR plots of HPAEC following exposure to S1 spike protein (25 nM) with and without HbA (*n* = 4). Representative OCR and ECAR plots were obtained from an individual set of experiment repeated three times. Student’s *t*-test unpaired, * *p* < 0.05 vs. respective untreated control.

**Figure 6 ijms-22-09041-f006:**
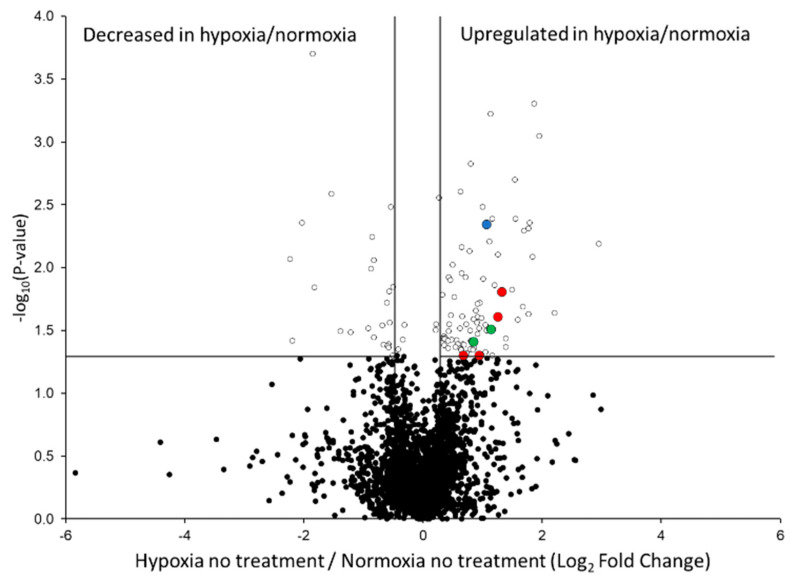
HIF signaling and ECM–receptor interaction pathway proteins were increased in the hypoxic/normoxic proteome. Data shown are untargeted µDIA proteomic quantification results from 2683 proteins (the 54 proteins that are listed in Appendix A but detected only in the hypoxic or normoxic conditions and are not included in this plot). ECM–receptor interaction pathway proteins are highlighted in red, the HIF signaling protein EGLN1 is highlighted in blue, and the P4HA1/P4HA2 proteins involved in altering the extracellular matrix are highlighted in green. All other proteins are shown as black-filled ovals (*p*-value > 0.05) and black circles with white centers (*p*-value < 0.05). The vertical lines denote fold-change values of −1.3 and +1.3 (without Log transformation), and the horizontal line represents a *p*-value = 0.05 (without Log transformation).

**Figure 7 ijms-22-09041-f007:**
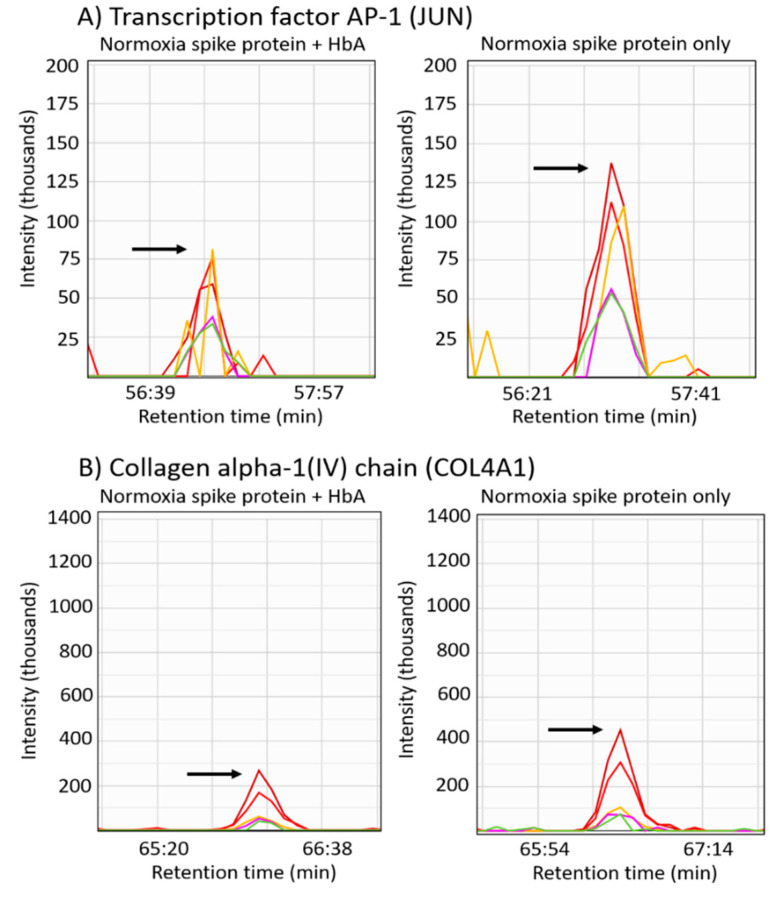
Proteins in the focal adhesion pathway were downregulated in normoxic S1 protein- and HbA-treated cells relative to S1 protein-only-treated cells. (**A**–**D**) Representative LC–MS/MS chromatograms are shown for b- and y-type sequencing fragment ions used to quantify peptides from each of the focal adhesion pathway downregulated proteins observed. The arrows in each chromatogram point to the peptide apex intensity. (**A**) The peptide AQNSELASTANMLREQVAQLK corresponding to the precursor *m*/*z* 768.07 (+3), (**B**) the peptide IVPLPGPPGAEGLPGSPGFPGPQGDR with precursor *m*/*z* 822.76 (+3), (**C**) the peptide TLYDFPGNDAEDLPFK with precursor *m*/*z* 921.43 (+2), and (**D**) shows the peptide DAQLSAPTK with precursor *m*/*z* 930.49 (+1).

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
