# Peer review of "Cell-Free Hemoglobin Does Not Attenuate the Effects of SARS-CoV-2 Spike Protein S1 Subunit in Pulmonary Endothelial Cells"

_ijms, 2021, doi:10.3390/ijms22169041_

Round 1

Reviewer 1 Report

The manuscript of Sirsendu Jana and coauthors is well-written and demonstrates the action of S1 Spike protein on lung endothelial cells. However,  some points should be improved to make the manuscript suitable for publication.

  1. Figure 2 requires the graph, based on presented Western blots. 
  2. According to Figure 5, the cell monolayer was not very confluent even in the control sample.  For the presented mitochondrial ROS production it is enough. However, since the level of confluence of endothelial monolayer is a gold indicator of endothelial barrier dysfunction, the presented control sample can disorient the reader. It would be very good to demonstrate the good confluent endothelial monolayer as control and damaged one as a result of S1 Spike protein. It could be done by any cytoskeleton or cell membrane fluorescent marker. 
  3. In a recently published paper in  AJP Lung titled "The SARS-CoV-2 Spike Protein Subunit S1 (S1SP) induces COVID-19-like acute lung injury in Κ18-hACE2 transgenic mice and barrier dysfunction in human endothelial cells" by Colunga Biancatelli etc, authors showed the concentration-dependent permeability and pulmonary barrier dysfunction followed Spike Protein Subunit 1 using human lung microvascular endothelial cells in ECIS. The authors should discuss this paper in comparison to their data.
  4. The addition of individual data points to the graphs would be beneficial. The error bars look large in Fig 4 and it would be better to visualize the variability with data points.

Author Response

Comments and Suggestions for Authors

The manuscript of Sirsendu Jana and coauthors is well-written and demonstrates the action of S1 Spike protein on lung endothelial cells. However, some points should be improved to make the manuscript suitable for publication.

  1. Figure 2 requires the graph, based on presented Western blots. 
  • We think the quantitation was asked for by this reviewer is for Figure 3 but not Figure 2. We have rearranged the blots and included the densitometric analysis in bar graphs in the revised manuscript.
  1. According to Figure 5, the cell monolayer was not very confluent even in the control sample.  For the presented mitochondrial ROS production, it is enough. However, since the level of confluence of endothelial monolayer is a gold indicator of endothelial barrier dysfunction, the presented control sample can disorient the reader. It would be very good to demonstrate the good confluent endothelial monolayer as control and damaged one as a result of S1 Spike protein. It could be done by any cytoskeleton or cell membrane fluorescent marker. 
  • We agree that the disruption of monolayer integrity is one of the best indicators of endothelial health, but for mitochondrial ROS production we found a diminished detection sensitivity with semi-confluent culture. In other words, for MitoSOX assay we always use lower density cells (within 50-60% confluency) to achieve a better sensitivity of the fluorescent probe. For the FITC-dextran passage experiment we cultured the cells until it became confluent to have a tight monolayer. We have modified the method section accordingly (Please see Results section 4.7).
  1. In a recently published paper in AJP Lung titled "The SARS-CoV-2 Spike Protein Subunit S1 (S1SP) induces COVID-19-like acute lung injury in Κ18-hACE2 transgenic mice and barrier dysfunction in human endothelial cells" by Colunga Biancatelli etc, authors showed the concentration-dependent permeability and pulmonary barrier dysfunction followed Spike Protein Subunit 1 using human lung microvascular endothelial cells in ECIS. The authors should discuss this paper in comparison to their data.

We thank the reviewer for pointing out this recent paper which is consistent with some of our findings. We have added the following in the text in section 2.2 around page 3 line 117 and added the suggested reference by Biancatteli as new reference 13 in the manuscript: 

“Due to endothelial barrier dysfunction that allows T-cell trafficking into the lung tissues autopsied from deceased COVID-19 patients [15] as well as acute lung injury reported in murine models of COVID-19 subjected to S1 protein [Biancatteli, et al new reference 16], we next assessed whether the S1 protein without all the remaining viral components is sufficient to cause endothelial barrier disruption by growing HPAEC in a tight monolayer and monitoring for changes in endothelial permeability by measuring the passage of FITC conjugated-dextran (40kD) through the monolayers [17,18]”

Additionally, we have made changes to the Discussion section around page 13 line 383:

“In subsequent stages of the infection process, immune cells are trafficked into the lungs and it has been suggested that this occurs in part due to a loss in pulmonary endothelial cell barrier function [15]. Therefore, we assessed the endothelial barrier function in the HPAEC analyzed in the presence of the S1 protein and found significantly higher cell permeability in cells treated with S1 protein compared to untreated control cells.  This finding confirms recent work by Biancatteli et al. [16] indicating the S1 protein alone is capable of inducing barrier dysfunction in human pulmonary microvascular cells and is notable given that HPAEC represent a substantial amount of all lung endothelial alveolar cells and provides a possible mechanistic basis for understanding why the lungs are a target organ for immune-mediated destruction in COVID-19 patients [35]. In addition to the endothelial cell barrier dysfunction, the process of infection also involves elevations in pro-inflammatory cytokines to such a degree it has been referred to as a “cytokine storm” with IL-6 serving as the upstream inducer of nuclear factor kappa B (NF-κB) and signal transducer and activator of transcription 3 (STAT3) [36]”

The addition of individual data points to the graphs would be beneficial. The error bars look large in Fig 4 and it would be better to visualize the variability with data points.

  • We have now replotted all the graphs to highlight individual data points.

Reviewer 2 Report

The manuscript discusses the results regarding the use of hemoglobin as a therapy for covid-related endothelial injury. The manuscript does not present positive results. However, this doesn't exclude the manuscript for consideration as these negative results could still be of use for other groups , exclude pathological pathways and stress the importance of an appropriate model and design for the ivnestigation of hemoglobin-related treatments. Despite the negative data, the manuscript is well written and the literature appropriate. Few comments:

  1. There are several puntuaction mistakes, the most common double spaces. Please provide a more throurough review of the manuscript.
  2. Figure legends should include statistics.
  3. Figure 2 A and B should have both the same maximal value for the X-axis (e.g. 20)
  4. Figure 3 would benefit for a bar graph in addition to the bands
  5. Figure 4: the significance of bars should be highlithed to their bar of comparison with brakets.
  6. The discussion is too verbose and out of focus. The authors should rewrite the section and spend more time on the limitations of the manuscript. Importantly, cells are cultures in media which has different properties from blood and as such, limits the effectiveness of hemoglobin treatments. Why authors have not considered using a in vivo animal model of covid exposure?
  7. Line 64 in the introduction: Critical care patients develop oxygen mismatch also due to mitochondirla dysfunction and bioenergetic failure. Authors should discuss this also as in relation to their negative results and focus.

Author Response

Comments and Suggestions for Authors

The manuscript discusses the results regarding the use of hemoglobin as a therapy for covid-related endothelial injury. The manuscript does not present positive results. However, this doesn't exclude the manuscript for consideration as these negative results could still be of use for other groups, exclude pathological pathways and stress the importance of an appropriate model and design for the investigation of hemoglobin-related treatments. Despite the negative data, the manuscript is well written and the literature appropriate. Few comments:

  1. There are several puntuaction mistakes, the most common double spaces. Please provide a more throurough review of the manuscript.

We have now corrected these punctuation/spacing mistake.

  1. Figure legends should include statistics.

-We changed the figure legends to include statistics.

  1. Figure 2 A and B should have both the same maximal value for the X-axis (e.g. 20)

We changed the figure as suggested by the reviewer.

  1. Figure 3 would benefit for a bar graph in addition to the bands
  • - We have rearranged the immunoblots and included all necessary densitometric analysis in bar graphs in the revised manuscript.
  1. Figure 4: the significance of bars should be highlighted to their bar of comparison with brakets.

- All the comparisons were made with respective untreated controls with P values <0.05. Since, they all were compared against the untreated controls we only used one asterisk sign (*) to indicate P<0.05 vs. corresponding controls.

  1. The discussion is too verbose and out of focus. The authors should rewrite the section and spend more time on the limitations of the manuscript. Importantly, cells are cultures in media which has different properties from blood and as such, limits the effectiveness of hemoglobin treatments. Why authors have not considered using a in vivo animal model of covid exposure?

We agree with this reviewer that in vitro and cell culture experiments have some limitations but are necessary precursors for our laboratory prior to performing more expensive in vivo studies with animals. We have indeed begun a collaborative investigation with colleagues within FDA (Office of Vaccine Research and Review) (OVRR) to look at the 3 elements of homeostasis (oxygen transport, oxygen sensing, and mitochondrial pathways) in hamster model of COVID19 infections [Selvaraji et al., Life Sciences Alliance http://doi.org/10.26508/lsa.202000886]. We are currently analyzing tissues lung tissue samples obtained from these animals, specifically we are looking mitochondrial function, ROS as well as a comprehensive proteomic analysis with some focus on HIF singling systems.  We have also revised the discussion of our manuscript to highlight the limitations of our study at the end of the Discussion section.

  1. Line 64 in the introduction: Critical care patients develop oxygen mismatch also due to mitochondirla dysfunction and bioenergetic failure. Authors should discuss this also as in relation to their negative results and focus.

We have in the introduction cited this observed disconnect between patient’s blood oxygenation and markers of hypoxia known as “happy hypoxia”. We cited a recent comprehensive review by Dhont et al., (Reference 7) describing the pathophysiology “happy hypoxia” in COVID-19.   Our observations are largely noted in cell culture experiments and we believe that the clinical consequences of this mismatch is beyond the scope of this paper.

Reviewer 3 Report

The manuscript is well written; however I have concerns outlined below. Please correct for unnecessary spaces throughout the text. For eg: Line 44, 57, 69, etc.

Specific concerns:

  • The authors should justify dose selection, 5-25nM, 24 hours.
  • Figure 1, the authors show that 5-25nM for 24 hours did not affect cell viability. This could be due to either using low dose or shorter incubation times?
  • Figure 2: Line 120, was it 12 hours or 24 hours? Figure 1 shows cell viability for 24 hours, why in figure 2, only 12 hours was selected? Please make Y axis scale consistent.
  • Figure 3, the authors must plot summarized data for western blot results.
  • Figure 4, Why hemoglobin treatment did not reverse the ICAM, VCAM, IL-1β but partially reverse CCL5 and CXCL10?
  • Line 203-204, a double space?
  • Figure 5, the authors should clarify what is basal OCR and maximal OCR?
  • Line 326, space between ways and In.
  • Figure 5, why hemoglobin treatment partially reversed glycolytic pathways?
  • What was purity of hemoglobin tested?
  • Suppliers of spike 1 protein should be mentioned.
  • The authors must cautiously report negative data because in most of the experiments, n=3-4 and internal variation is very high; thus, data interpretation is critical. I suggest increasing “n” number to gain confidence in data. Was SD used instead of SEM?
  • Is the concentration of spike 1 protein enough to elicit cell signaling, does the concentration used in this study correspond to concentrations in vivo?
  • Very limited discussion about HIF-1α.
  • The authors report that spike protein causes endothelial damage, cytokine release and ROS. The mechanism should be clearly described or speculated.
  • Can authors comment on spike protein 2? What is its part in initiating signaling?

Author Response

Comments and Suggestions for Authors

The manuscript is well written; however I have concerns outlined below. Please correct for unnecessary spaces throughout the text. For eg: Line 44, 57, 69, etc.

Spaces throughout the text were corrected. We thank the reviewer for pointing out this error.

Specific concerns:

  • The authors should justify dose selection, 5-25nM, 24 hours.

-For dose selection we have followed the literature available where in vitro experiments were done with spike S1 protein with cultured cells especially with endothelial cells. Some of those studies used much lower or similar concentrations whereas some used higher concentrations. Therefore, we tested the spike protein S1 between an intermediate range (5-25 nM) which corresponds to approximately 0.4 -2.0 µg/ml respectively. We have included the references at the beginning of the Results section 2.1

Figure 1, the authors show that 5-25nM for 24 hours did not affect cell viability. This could be due to either using low dose or shorter incubation times?

-As we have stated earlier that the dose we used correspond to 2.0 µg/ml which is a very high concentration compared to what is achieved by vaccination. Although there is no clear data available regarding the actual spike protein concentrations in COVID 19 patients.  Our data is also consistent with prior studies indicating the S1 protein does not impact cell viability, for example see:  Buzhdygan et al. [Reference 12]  

  • Figure 2: Line 120, was it 12 hours or 24 hours? Figure 1 shows cell viability for 24 hours, why in figure 2, only 12 hours was selected? Please make Y axis scale consistent.

-We thank the reviewer for catching this error. All the results presented in this report are from 12 hours exposure. However, we also tested 24 h for cell viability assays (data not shown). Our 24h data did not differ from 12 h data in terms of cell viability. We have changed the text accordingly.

  • Figure 3, the authors must plot summarized data for western blot results.

- We have rearranged the immunoblots and included the bar graphs for Western blot results in Figure 3.

  • Figure 4, Why hemoglobin treatment did not reverse the ICAM, VCAM, IL-1β but partially reverse CCL5 and CXCL10?

We don’t know the basis of this, but Hb/heme are an important immunostimulating agent and oxidative factor contributing to endothelial cell activation (F A Wagener et al., Proc Soc Exp Biol Med. 1997 Dec;216(3):456-63). Although the trend was lower in S1+HbA group than the S1 group however, we would like to draw the reviewer’s attention that we did not see any significant difference between those groups. We have now replotted the results.

Line 203-204, a double space?

-we corrected the error.

  • Figure 5, the authors should clarify what is basal OCR and maximal OCR?

-We have described the basal OCR and maximal OCR calculations in the methods section. Additionally, we now also included a short description of the parameters in the results section.

  • Line 326, space between ways and In.

-we corrected the error.

  • Figure 5, why hemoglobin treatment partially reversed glycolytic pathways?

It is very difficult to explain the impact of Hb on glycolytic pathways in this simple cell culture system, but we did show that Hb through its interaction with band3 may influence glycolytic pathways (Jana et al., JCI Insight. 2018 Nov 2;3(21):e120451; Strader MB, et al., Sci Rep. 2020 Aug 26;10(1):14218). Further, the rise in glycolytic reserve capacity in the endothelial cells by S1 protein can be an indirect compensatory effect of subtle mitochondrial dysfunction what we were not able to capture by XF assay but was evident from the rise in mitochondrial ROS. This is particularly true in endothelium as these cells are heavily dependent on the glycolysis as their energy source. Since, Hb can act as a mild uncoupler (as we have seen before) and noticeable in Figure 5E causing a slight rise in the maximal OCR, the partial reversal of glycolytic capacity can be another indirect effect of Hb addition.

  • What was purity of hemoglobin tested?

The purity of Hb was indeed tested using IEF and HPLC. More importantly endotoxin levels were kept to a minimum as we have already indicated in the material and methods under preparation of hemoglobin from blood.

  • Suppliers of spike 1 protein should be mentioned.
  • We have indicated the source of spike protein in the methods section.
  • The authors must cautiously report negative data because in most of the experiments, n=3-4 and internal variation is very high; thus, data interpretation is critical. I suggest increasing “n” number to gain confidence in data. Was SD used instead of SEM?

-We ran some pilot experiments prior to main experiments, the results were similar, but those results were not included in this manuscript. Additionally, we also checked the Hb interaction with spike protein S1 in vitro using another endothelial cell line (HUVEC) and found similar changes as seen with HPAEC. All error bars represent SD not SEM.

  • Is the concentration of spike 1 protein enough to elicit cell signaling, does the concentration used in this study correspond to concentrations in vivo?

-Yes, the concentration of the S1 protein was sufficient to induce cell signaling, particularly in normoxic cells where our proteomics analysis indicated proteins were increased in the lysosome, ECM-receptor interaction, and focal adhesion pathways.  However, in the hypoxic condition the HPAECs were protected from there being many changes in protein differences.  The finding of few protein differences in the hypoxic cells is consistent with the reduction in ACE-2 in hypoxia shown in the Western blot in Figure 3.  Regarding the second part of the question as to whether the concentrations of S1 protein relate to those found in patients, we have not found a literature reference to answer this very good question.  However, the concentrations we used in our manuscript for in vitro analyses were based on other cell culture studies with the S1 protein. 

Very limited discussion about HIF-1α.

-HIF-1 α was discussed extensively in in our manuscript from the introduction to the discussion sections and numerous references were included.  As a result, we believe that HIF-1α was discussed adequately throughout the text of the paper (introduction, results, and discussion) as well as very updated literature on the role of HIF-1α in COVID-19 infection was provided.

  • The authors report that spike protein causes endothelial damage, cytokine release and ROS. The mechanism should be clearly described or speculated.

-We have now discussed the possible mechanism more elaborately in the discussion section

  • Can authors comment on spike protein 2? What is its part in initiating signaling? 
  • The role of spike protein subunit 2 on endothelial cells was studied by another group (Buzhdygan et al, Ref 12) and we have discussed their findings in the discussion section. The results by Buzhdygan et al. and by few other groups showed no or much less toxicity on cultured cells.

Round 2

Reviewer 1 Report

The manuscript by Jana etc was significantly improved and could be accepted in the present form.

Author Response

We are extremely thankful to the reviewer for the positive comments.

Reviewer 2 Report

No more comments

Author Response

We are extremely thankful to the reviewer.

Reviewer 3 Report

The manuscript is greatly improved, however, "n" number should be increased if stable cell culture model is established.

Author Response

Comments and Suggestions for Authors

The manuscript is greatly improved, however, "n" number should be increased if stable cell culture model is established.

We are extremely thankful to the reviewer for the positive comments.

We understand the reviewer’s concern regarding the variability issues commonly encountered with cell culture based experiments as different cell line-based models have many limitations due to variations in observed responses. That was also our primary concern. So, we ran similar set of experiments with spike protein S1 in vitro using another endothelial cell line (HUVEC) and found similar changes as we have stated earlier. However, we want to draw kind attention to the reviewer that we now increased the ‘n’ number to 4 for most of the experiments and replotted the results with new P values indicated on top of each bar diagrams. Although there are slight variations in the mean values but overall the trend is similar to the first version results. We are also doing similar experiments with S1 protein as well as with the heat inactivated SARS virus using the same cell line  and also with an animal model using Syrian hamsters with our collaborators at the Center for Biologics Evaluation and Research, FDA (Dr. Tony Wang PhD, OVRR/CBER/FDA; https://www.fda.gov/vaccines-blood-biologics/biologics-research-projects/vaccines-research). So far, our results are very consistent.